# Novel *Pseudomonas* Species Prevent the Growth of the Phytopathogenic Fungus *Aspergillus flavus*

**DOI:** 10.3390/biotech13020008

**Published:** 2024-03-30

**Authors:** Franciene Rabiço, Tiago Cabral Borelli, Robson Carlos Alnoch, Maria de Lourdes Teixeira de Moraes Polizeli, Ricardo R. da Silva, Rafael Silva-Rocha, María-Eugenia Guazzaroni

**Affiliations:** 1Department of Cell and Molecular Biology, Faculdade de Medicina de Ribeirão Preto, University of São Paulo, Av. Bandeirantes, 3.900, Ribeirão Preto 14040-901, SP, Brazil; franciene.oliveira@usp.br (F.R.); tiago.borelli@usp.br (T.C.B.); 2Department of Biology, Faculdade de Filosofia, Ciências e Letras de Ribeirão Preto, University of São Paulo, Av. Bandeirantes, 3.900, Ribeirão Preto 14040-901, SP, Brazil; robsonalnoch@usp.br (R.C.A.); polizeli@ffclrp.usp.br (M.d.L.T.d.M.P.); 3Department of Biomolecular Sciences, Faculdade de Ciências Farmacêuticas de Ribeirão Preto, University of São Paulo, Av. Bandeirantes, 3.900, Ribeirão Preto 14040-901, SP, Brazil; ridasilva@usp.br; 4ByMyCell Inova Simples, Av. Dra. Nadir Aguiar, 1805, Ribeirão Preto 14056-680, SP, Brazil; rafael@bymycell.com

**Keywords:** bacteria volatile compounds, *Pseudomonas*, biological control, phytopathogenic fungus, microbial antagonistic activity

## Abstract

In response to the escalating demand for sustainable agricultural methodologies, the utilization of microbial volatile organic compounds (VOCs) as antagonists against phytopathogens has emerged as a viable eco-friendly alternative. Microbial volatiles exhibit rapid diffusion rates, facilitating prompt chemical interactions. Moreover, microorganisms possess the capacity to emit volatiles constitutively, as well as in response to biological interactions and environmental stimuli. In addition to volatile compounds, these bacteria demonstrate the ability to produce soluble metabolites with antifungal properties, such as APE Vf, pyoverdin, and fragin. In this study, we identified two *Pseudomonas* strains (BJa3 and MCal1) capable of inhibiting the in vitro mycelial growth of the phytopathogenic fungus *Aspergillus flavus*, which serves as the causal agent of diseases in sugarcane and maize. Utilizing GC/MS analysis, we detected 47 distinct VOCs which were produced by these bacterial strains. Notably, certain volatile compounds, including 1-heptoxydecane and tridecan-2-one, emerged as primary candidates for inhibiting fungal growth. These compounds belong to essential chemical classes previously documented for their antifungal activity, while others represent novel molecules. Furthermore, examination via confocal microscopy unveiled significant morphological alterations, particularly in the cell wall, of mycelia exposed to VOCs emitted by both *Pseudomonas* species. These findings underscore the potential of the identified BJa3 and MCal1 *Pseudomonas* strains as promising agents for fungal biocontrol in agricultural crops.

## 1. Introduction

Fungi belonging to the genus *Aspergillus* have historically been utilized as biological agents for controlling the gray sugarcane mealybug *Pseudococcus sacchari*, a significant pest affecting sugarcane crops worldwide [1]. However, this approach was developed previously to the discovery of aflatoxins, hepatocarcinogenic secondary metabolites produced by fungi, in the early 1960s [2,3] (Coomes and Sanders, 1963; Hartley et al., 1963. Aflatoxins represent pervasive contaminants in food and feed globally, primarily affecting consumers through the ingestion of contaminated seeds and edible plant parts. *Aspergillus flavus*, known for its production of aflatoxin types B and G, has been identified in maize and traditional herbal products such as yerba mate [4,5]. Moreover, it stands as the predominant phytopathogenic species affecting sugarcane and its by-products in Brazil [6]. Fungal proliferation in crop plants not only incurs substantial economic losses but also poses significant health risks [7,8,9]. Consequently, researchers have endeavored to develop effective strategies for mitigating fungal growth, both pre- and post-harvest [8,9]. Biological control methods have emerged as a particularly prominent avenue due to their cost-effectiveness and environmental compatibility [8]. Estimates suggest that fungal diseases contribute to approximately 30% of global crop losses [1]. Various microorganisms offer potential for suppressing plant diseases by enhancing host immunity or directly inhibiting pathogens [3]. Accordingly, there is a pressing need to explore plant growth-promoting bacteria (PGPB), also known as multifunctional bacteria, which serve as antagonists against phytopathogens, offering an eco-friendly alternative to conventional agrochemicals. PGPBs employ diverse mechanisms to promote plant health and growth, among which the production of volatile organic compounds (VOCs) holds particular significance. VOCs, derived from various biochemical pathways, possess low boiling points and high vapor pressures [10], rendering them potent signaling molecules with potential applications in disease management and plant stimulation.

Volatile organic compounds (VOCs) comprise a diverse array of metabolites that exhibit volatility and are synthesized by various microorganisms. This class of compounds includes alcohols, ketones, thioesters, phenols, and benzene derivatives [11]. Notably, VOCs offer significant advantages as they are biodegradable and can be employed as gaseous treatments, thereby mitigating the accumulation of toxic residues on fruits [12,13,14]. Recent years have seen the growth of interest in bacterial VOCs due to their essential role as signaling molecules in the biocontrol of phytopathogens and the stimulation of plant growth. The gaseous nature of VOCs facilitates their efficient dispersion through porous soil, enabling them to traverse considerable distances. Consequently, VOCs emerge as promising candidates for biocontrol strategies, bolstering the efficacy of antagonistic mechanisms against specific microorganisms [15,16].

Among VOC-producing microorganisms, the bacterial genus *Pseudomonas* has emerged as a promising biofumigation agent for combatting various fungal diseases. Studies have demonstrated its efficacy in controlling gray mold caused by *Botrytis cinerea* in *Medicago truncatula* plants [17,18] and apples [19], as well as tomato wilt induced by *Ralstonia solanacearum* [20], and blue mold affecting apple and citrus fruits caused by *Penicillium expansum* and *Penicillium italicum* [21]. These findings underscore the potential of Pseudomonas-derived VOCs as an appealing alternative for postharvest disease management strategies.

In addition to volatile organic compounds (VOCs), attention should also be directed towards diffusible and soluble compounds, as they cover a vast array of potentially biologically active substances capable of inhibiting the growth of phytopathogens [22,23,24,25]. Among these compounds are cyclic lipopeptides belonging to the iturin, surfactin, and fengicin families, which are renowned for their potent activity against a broad spectrum of phytopathogens, including bacteria, fungi, oomycetes, and viruses [26,27,28].

In the present study, we identified two soil *Pseudomonas* species (BJa3 and MCal1) that emit relevant volatile signals that were able to inhibit the mycelial growth of *A. flavus*. Our findings reveal that these bacterial isolates produce volatile organic compounds (VOCs) which exhibit potent antagonistic activity when co-cultivated with the fungus. To characterize the VOCs emitted by our *Pseudomonas* isolates during co-cultivation with *A. flavus* and in isolation, we employed solid-phase microextraction coupled with gas chromatography–mass spectrometry (SPME/GC–MS) analysis. Furthermore, confocal laser scanning microscopy (CLSM) analysis was conducted to ascertain the damage inflicted by the bacterial VOCs on the hyphae, mycelia, and conidia of *A. flavus*, thereby impeding its development. Additionally, genomic and experimental analyses indicated that the bacterial isolates produce soluble and diffusible compounds capable of inhibiting the mycelial growth of *A. flavus*. Overall, our in vitro and in silico results underscore the multifaceted antimicrobial activity exhibited by strains BJa3 and MCal1 at both the diffusible and volatile levels, thus underscoring their potential utility as valuable fungicidal agents.

## 2. Materials and Methods

### 2.1. Microorganisms and Growth Conditions

The phytopathogenic fungus *A. flavus* (URM 7262) was kindly provided by the Collection of cultures—Micoteca URM, from the Biological Sciences Center of the Federal University of Pernambuco. Fungus was cultivated on potato dextrose agar (PDA; agar, 15 g/L; dextrose, 20 g/L; potato extract, 4 g/L) (Sigma-Aldrich, Burlington, MA, USA) for VOC assays and on Vogel Medium (Na_3_C_6_H_5_O_7_. 2H_2_O, 125 g/L; KH_2_PO_4_, 250 g/L; NH_4_NO_3_, 100 g/L; MgSO_4_. 7H_2_O, 10 g/L; CaCl_2_. 2H_2_O, 5 g/L; Trace element solution, 5 mL/L; biotin solution, 2.5 mL/L; sucrose, 15 g/L) for diffusion assay. The 10 evaluated bacteria (all of which belong to the *Pseudomonas* genus) were isolated from soil and sugarcane juice samples in the region of Ribeirão Preto, Brazil (21°10′40″ S; 47°48′36″ W). Bacteria were isolated using abundant and inexpensive carbon sources such as sugarcane juice (MCal1, M3 and MGal98 strains) and sodium benzoate (BJa3 and Bagro211 strains). After, all bacteria were routinely grown on Luria–Bertani (LB; NaCl, 10 g/L; Tryptone, 10 g/L; Yeast Extract, 5 g/L) (Sigma-Aldrich, Burlington, MA, USA) medium at 30 °C.

### 2.2. Phylogenetic Analysis of BJa3 and MCal1 Isolates

Total DNA of *Pseudomonas* spp. selected strains was extracted using the GeneJET Genomic Purification Kit (Thermo Fischer Scientific, Waltham, MA, USA) according to the manufacturer’s instructions. The concentration of DNA was determined fluorometrically using the Qubit^®^ 3.0 (Qubit^®^ dsDNA Broad Range Assay Kit, Life Technologies, Carlsbad, CA, USA). DNA library was prepared using the Nextera XT DNA Library Prep Kit (Illumina, San Diego, CA, USA), assessed for quality using the 2100 Bioanalyzer (Agilent Genomics, Santa Clara, CA, USA), and subsequently submitted to sequencing using the Illumina HiSeq (2 × 150 bp) platform (Illumina, San Diego, CA, USA). The phylogenetic analysis of the evaluated isolates was performed using the whole genome. To assemble the genomes, the sequencing quality was first analyzed using the FastQC tool (v. 0.11.9) followed by trimming with the Trimmomatic program (version 0.38.0) [29] and the removal of possible human contamination reads with the bowtie2 tool (v. 2.3.5.1) [30]. The SPAdes assembler (v. 3.12) [31] was chosen for genome assembly and the MEGAHIT tool (v. 1.2.9) [32] was used as a secondary assembler. Finally, the FGAP (v. 7.17) [33], which consists of a tool to fill in the undefined bases of assemblies, was used with the extended paired-end bases. Additionally, the annotation of the genes was carried out with the PROKKA tool (v. 1.12) [34]. The phylogenetic identification of the isolates was performed using the TYGS platform [35], and the classification was based on the Digital DNA-DNA hybridization (dDDH) value [36].

### 2.3. Detection of VOCs Antagonistic Activity In Vitro

The impact of VOCs produced by the 10 bacteria from various genera was assessed for the initial screening using 2-compartment Petri dishes. *A. flavus* was raised in a Petri dish filled with PDA medium and incubated for five days in a growth chamber at 28 °C. One compartment of the plate containing LB medium was spread with 100 μL of a fresh bacterial suspension (about 5 × 10^8^ CFU), and the other compartment containing PDA was dripped with a spore solution containing 10^6^ spores mL^−1^. The plates were covered with Parafilm^®^ and kept at 28 °C for two days of incubation. Plates without bacterial inoculum (negative control) or containing bacteria *E. coli* DH5α and *P. putida* KT 2440 were used as experiment controls.

The 5 bacteria that had the more significant visual effects on the growth of *A. flavus* from an initial screening were then tested in triplicate. Then, the top 2 isolates in terms of fungal inhibition were selected for ensuing examinations.

For VOCs synergy tests on fungal growth, the 2 most promising isolates were placed on the same plate and cultivated jointly. Each bacterial inoculum was spread out over 50 μL (about 5 × 10^8^ CFU) in one of the divided plate compartments. A solution containing roughly 10^6^ *A. flavus* spores was inoculated on the opposing side. Plates without bacterial inoculum (negative control) or containing bacteria *E. coli* DH5α and *P. putida* KT 2440 were used as experiment controls.

### 2.4. Detection of Soluble Antagonist Compounds

To detect the presence of antifungal compounds in bacterial cultures cultivated in liquid medium, strains BJa3, MCal1, and DH5α were cultivated in 50 mL of LB medium at 30 °C and 200 rpm for 16 h. *A. flavus* fungus was cultivated in 50 mL of Vogel medium at 28 °C and 150 rpm during 48 h. Then, inocula were collected and mixed at a ratio of 1/10 (100 μL of supernatant of *A. flavus* to 900 μL of supernatant of bacterial isolates) and incubated again in LB medium at 30 °C and 200 rpm for 16 h. Inocula containing only bacteria were also made. After this time, the inocula were collected and centrifuged at 10,000 rpm for 10 min, and supernatants were filtered through a 0.22 mm filter. For the negative control, Vogel, Vogel plus LB, and LB media were used instead of cell-free supernatant. One mL of the supernatant was mixed with agar–LB and poured into a Petri dish. A 5 μL solution of *A. flavus* was inoculated in the center of each plate and incubated for 2 days at 28 °C. Later, mycelial growth was measured. Three independent assays were performed. In addition, the online tool antiSMASH (https://antismash.secondarymetabolites.org/#!/start) (accessed on 24 May 2023) [12] was used to determine which soluble compounds were produced by the BJa3 and MCal1 bacteria, using the assembled genomes of the respective bacteria as input.

### 2.5. Statistical Data Analysis

In order to determine if there were significant differences between the 2 groups, one-way ANOVA test was used. Tukey’s test was used to compare means and standard deviations using GraphPad Prism v.8.0.2 software (San Diego, CA, USA). *p* values less than 0.05 were considered statistically significant.

### 2.6. Morphological Characterization of A. flavus after VOC Exposure of BJa3, MCal1, and DH5α by Confocal Laser Scanning Microscopy (CLSM)

Changes in the external morphology of *A. flavus* after exposure to the selected isolates, BJa3, MCal1, and DH5α, at a final concentration of 5 × 10^8^ CFU of bacterial solution were examined using CLSM. DH5α strain as well as microcultures not exposed to bacterial cultures were used as negative controls. Fungus were cultivated, with minor modifications, according to the protocol by Li et al. (2010) [37]. *A. flavus* fungus was collected after 2 days of growth using an inoculation loop to preserve the integrity of the mycelia and placed at the four corners of a PDA-medium rectangle. This structure was placed on a microscope slide and inserted into a 50 mL conical flask. Previously, cultures of the selected bacteria in 5 mL of LB medium were cultivated inside the flask. This system was incubated for 2 days at 28 °C. Later, the samples were washed in Phosphate-Buffered Saline (PBS) and stained with 50 μL of calcofluor white (Sigma-Aldrich, USA) for 10 min. Images were acquired in Confocal Microscopy Multiuser Laboratory (LMMC—USP) with a Leica SP5 microscope (Leica Microsystems, Wetzlar, Germany) using plan apochromatic 40× (NA 1.25, oil) and 63× (NA 1.4, oil). Calcofluor-stained samples were recorded in blue channel (395 nm) and the emission wavelength was 440 nm.

### 2.7. Bacterial VOCs Identification by Solid-Phase Microextraction Coupled to Gas Chromatography-Mass Spectrometry (SPME/GC–MS)

Volatilome analysis of the antagonists was performed at the Organic Micromolecule Mass Spectrometry Center (CEMMO) located at FCFRP/USP (Ribeirão Preto, SP, Brazil). Fifty microliters of bacterial suspension (~10^8^ CFU) of the selected isolates (BJa3, MCal1 and DH5α) and of the *A. flavus* fungus were inoculated in 10 mL flasks with screw caps and PTFE/Silicone septa (Supelco, Bellefonte, PA, USA), and 1 mL of the respective culture medium was deposited on each side of the flask (LB for bacteria and PDA for fungus). Microorganisms were incubated for 48 h at 28 °C in quadruplicates. For control samples, a flask containing only culture medium without bacterial inoculum was used as well as *E. coli* DH5α.

VOCs produced were collected by using the solid-phase microextraction (SPME) technique [38] with a StableFlex fiber 2 cm of 50/30 μm divinylbenzene/carboxen/polydimethylsiloxane (Supelco, Bellefonte, PA, USA). The SPME fiber was inserted into the 10 mL glass vials equipped with screw caps and PTFE/Silicone septa and extraction was performed in headspace mode at 55 °C for 30 min. Then, the fiber was desorbed into the GC port at 240 °C for 3 min in split mode and separated in a GC-MS QP2010 Ultra (Shimadzu, Kyoto, Japan) equipped with a ZB-5MS column (30 m × 0.25 mm inner diameter × 0.25 µm film thickness). The temperature program was from 40 °C (held for 3 min) to 220 °C at 7 °C/min and then to 260 °C at 12 °C/min (held for 2 min). The mass spectrometer was operated in the electron ionization mode at 70 eV. The ion source and transfer line temperature were set, respectively, at 250 °C and 260 °C. Mass spectra were scanned in the range *m*/*z* 35–400 amu. For VOCs identification, mass spectra were compared with those of the Global Natural Products Social Molecular Networking (GNPS) mass spectrometry library [39].

### 2.8. Bacterial Volatilome Analysis

The Shimadzu GGD files were converted to CDF format with Openchrom software (v. 15.0) (https://lablicate.com/platform/openchrom) (accessed on 7 July 2023). The converted spectra were uploaded to the GNPS platform [15] and pre-processing was performed with MSHUB-GC workflow (https://gnps.ucsd.edu/ProteoSAFe/status.jsp?task=9532209f010b442a9efe95a7897efd66) (accessed on 7 July 2023). The deconvoluted fragmentation spectra obtained from pre-processing were submitted to the MOLECULAR-LIBRARY SEARCH-GC workflow (https://gnps.ucsd.edu/ProteoSAFe/status.jsp?task=b6d863f7266f4bb7a0918e576abe64fb) (accessed on 7 July 2023) to perform spectral networking and automated spectral matching, which was where the compound annotations were obtained from. Principal Component Analysis was performed after spectra total peak area normalization and auto-scaling. Heatmaps were plotted using the mean of the spectra peak area relative to the abundance of each group (*A. flavus* alone; *A. flavus* + bacteria; bacteria alone). We searched for statistically significant differences in spectra peak area in each group using the Kruskal–Wallis test [40] (α = 0.05).

## 3. Results and Discussion

### 3.1. BJa3 and MCal1 Produce Volatile Compounds with Antifungal Activity

Numerous species within the genus *Pseudomonas* are known inhabitants of soil and rhizosphere environments, and concurrently display antagonistic behaviors against phytopathogenic fungi [41,42,43,44,45,46]. Motivated by these findings, we investigated *Pseudomonas* spp. isolates sourced from soil and sugarcane juice as potential candidates for biocontrol agents. Initially, we conducted a screening of 10 bacterial isolates to identify VOC-producing strains with inhibitory effects against the growth of *A. flavus*. Based on quantitative assessments of mycelium diameter (Figure 1), we identified the top five most promising bacteria. *Escherichia coli* DH5α and the soil bacterium *Pseudomonas putida* KT 2440 were employed as controls. As depicted in Figure 1, the strains BJa3 and MCal1 exhibited approximately 50% inhibition of *A. flavus* growth, followed by M3 at 40%, relative to the control (without bacterial inoculum). Notably, cultivation of *A. flavus* in the presence of strain MCal1 resulted in conspicuous alterations to the mycelium’s appearance, characterized by discernible differences in color and structure compared to the negative control (Appendix A). In contrast, *E. coli* DH5α and *P. putida* KT 2440 did not exhibit an inhibition of fungal growth. Consistent with our observations, *E. coli* DH5α previously demonstrated a lack of antifungal activity in the literature [20]. Subsequently, the isolates which demonstrated the most potent antifungal activity (BJa3 and MCal1) were selected for further experimentation. The measurements of the halos were carried out after two days of incubation, because after this period, the supply of nutrients for the bacteria becomes sparser, favoring a drop in their growth and, consequently, a lower production of VOCs.

A variety of factors affect the composition of the volatile metabolome subset, also known as the volatilome. These factors include elements such as culture media, temperature, pH, interactions with other organisms, and growth stage [45,47]. Remarkably, prior investigations have revealed that nutrient-rich media, such as Luria–Bertani (LB), not only promote bacterial growth and fungal inhibition but also enhance the production of volatile organic compounds (VOCs) by bacteria [48]. Moreover, distinct volatile compositions have been observed between monocultures and microbial mixtures [48]. To evaluate the potential synergistic antagonistic effect of VOC emissions against *A. flavus*, we conducted co-cultivation experiments involving the two most promising bacteria, BJa3 and MCal1, in conjunction with the fungus. Contrary to our expectations, the co-cultivation of the BJa3 and MCal1 strains did not result in a significant inhibition of fungal growth (Figure 1). Recent investigations have elucidated that interactions among different bacterial species can either induce or suppress the emission of volatile chemicals that modulate microbial growth dynamics [49,50]. These findings highlighted a predisposition for greater antifungal activity when antagonistic bacteria are cultivated in nutrient-rich media and separately.

Despite the demonstrated efficacy of VOCs in defending against phytopathogens in controlled laboratory assays, their effectiveness in enhancing plant defenses demands a range of additional tests that include various factors that must be considered. Methodologies for the direct application of VOCs in agricultural fields and greenhouses are rarely explored and require further investigation. Only a few studies have tested the effect of VOCs on plant growth promotion and disease suppression under field conditions through soil-drench methods, spray applications, and seed treatments [51,52,53,54].

Several factors may cause differences between in vitro and in vivo tests. One of these is the fact that laboratory experiments demonstrating the efficacy of VOCs typically involve much higher bacterial concentrations than those achievable in open fields. Additionally, while the high biodegradability of VOCs may minimize their negative environmental impact, the reactivity of VOCs limits the distance they travel in fields and barns, thereby limiting their persistence and activity [55,56,57]. Moreover, the very essence of VOCs (volatility) means that these compounds are highly influenced by meteorological factors such as wind speed and direction, humidity, rainfall, and temperature.

Another factor that should be considered is aeration conditions. Under aerobic conditions, bacteria utilize any carbon source for cell growth, with only a small fraction being used for VOC production. Under microaerophilic and anaerobic conditions, bacteria initiate fermentation by utilizing carbon sources, and these are involved in the biosynthesis and emission of various VOCs [58,59].

Despite all the challenges in field conditions, VOCs can still be effectively utilized, as the necessary concentration of VOCs can be maintained in enclosed environments [59]. However, to fully assess their effects, multiple analyses and investigations are required to examine the efficacy of VOCs produced under natural conditions during plant–microorganism interactions.

### 3.2. The Strain MCal1 Represents a Novel Species within the Pseudomonas Genus, while BJa3 Is Categorized within the Pseudomonas Soli Species

Based on the antifungal activity screening results, we elected to take a deeper look into the genomic characteristics of the strains MCal1 and BJa3. These strains, isolated from soil (BJa3) and sugarcane juice (MCal1), underwent genome sequencing. Digital DNA-DNA hybridization (dDDH) analysis revealed their affiliation with the genus *Pseudomonas* (Figure 2). Specifically, the BJa3 strain exhibited a close similarity to *Pseudomonas soli* species, with a dDDH4 value exceeding 90%, indicating that they very possibly belong to the same species (Appendix A and Figure 2A) [36]. Conversely, the *Pseudomonas* sp. MCal1 strain clustered within the same monophyletic group as *Pseudomonas glycinae* and *Pseudomonas gozinkensis* (refer to Figure 2B). However, upon comparing the MCal1 genome with 14 closely related *Pseudomonas* species genomes available on GenBank, the dDDH4 value surpassed 50% only with the two most similar strains (Appendix A). This result suggests that the *Pseudomonas* sp. MCal1 may be a new *Pseudomonas* species.

The genome size of BJA3 was estimated to be 5.9 Mb with 5272 coding sequences and the GC content was estimated to be 64.05% (Appendix A). The MCal1 strain has a genome size of 6.2 Mb, 5650 coding sequences and a GC percentage of 60.42%. Both DNA GC content and genome size can be considered as important taxonomic markers. It was observed that, in *Pseudomonas*, the GC content varies from 48.3% to 68.3% [60,61]. Genome size differences vary within the genus, as does the corresponding gene content, and, in this case, genome size is directly related to the number of coding sequences. In *Pseudomonas*, coding genes ranged from 2803 (genome size 3.03 Mb) to 6895 (genome size 7.38 Mb) [62]. Therefore, both the genome size and coding sequences and the GC content of the sequenced genomes are within the expected ranges of bacteria belonging to the *Pseudomonas* genus.

**Figure 2 biotech-13-00008-f002:**
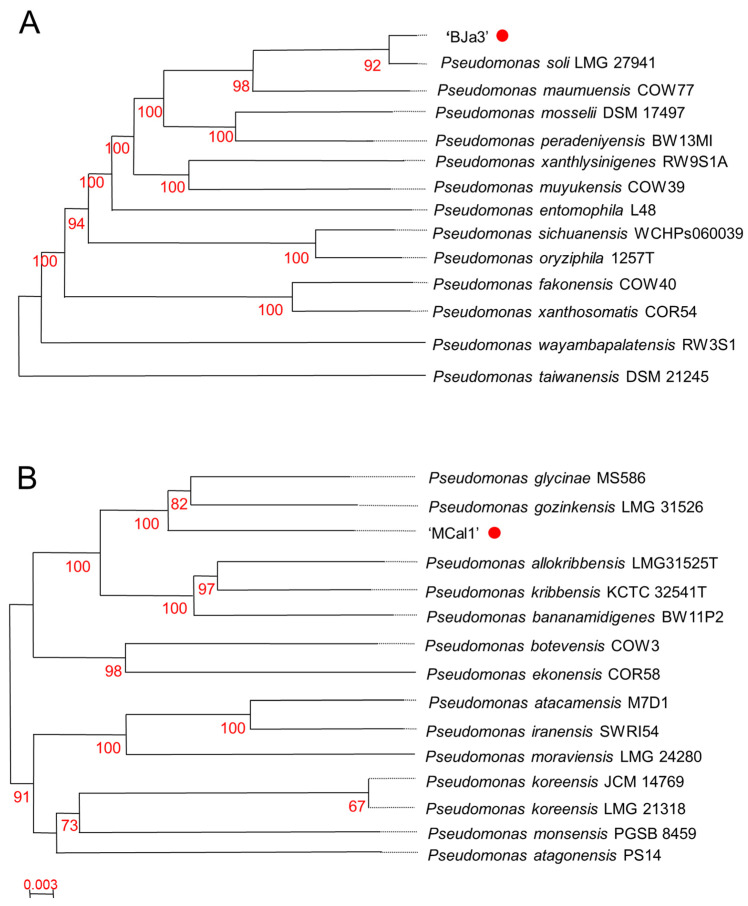
Taxonomic affiliation of *Pseudomonas* sp. strains BJa3 and MCal1. Tree of BJa3 (**A**) and MCal1 (**B**) inferred with FastME [63] from Genome BLAST Distance Phylogeny method (GBDP) distances calculated from genome sequences. Branch lengths are scaled in terms of GBDP distance formula d5; numbers above branches are GBDP pseudo-bootstrap support values from 100 replications. The trees were constructed using the TYGS tool.

### 3.3. MCal1 and BJa3 Also Exhibit Antimicrobial Effects on A. flavus through Diffusible Compounds Production

Considering that the strains BJa3 and MCal1 were able to hamper fungal growth in two-compartment Petri dishes with VOCs, we wondered if antagonistic activity was also produced by agar-diffusible compounds. It has been shown that some compounds that are soluble in liquid media are capable of inhibiting fungal development and altering some characteristics of the mycelium [64]. To check whether our bacterial strains produced soluble compounds of interest, we first carried out a search for secondary metabolites using the whole genomes of these bacteria as inputs to the antiSMASH tool (v. 7.1.0) (Appendix A). 

The genomic results showed shared antifungal secondary metabolites between strains BJa3 and MCal1, notably APE Vf, pyoverdin, and fragin. These compounds exhibit distinct mechanisms of action. The arylpolyene gene cluster (APE Vf) identified in the genome of strain BJa3 represents a compound with multifaceted benefits, contributing to plant protection against oxidative stress and exhibiting various biological functions. This includes effective biocontrol against the pathogenic fungus Fusarium oxysporum, thereby enhancing plant survival [65,66].

Another crucial identified gene cluster involves the siderophore pyoverdin, which plays a significant role in impeding biofilm formation on the phytopathogenic fungus *Aspergillus fumigatus* [67,68]. Additionally, fragin, categorized as a metallophore, demonstrates noteworthy antifungal activity primarily based on its ability to chelate metals, thereby disrupting fungal growth [69].

Several noteworthy secondary metabolites were exclusively identified within the BJa3 strain. One such compound, putisolvin, stands out as a lipopeptide, functioning as a biosurfactant. Its significance lies in facilitating swarming motility and biofilm formation while exhibiting zoosporicidal and antifungal activities [70]. Another compelling cluster identified at the genome level was pseudopyronine A/pseudopyronine B, demonstrating potent activity against plant pathogenic fungi [71]. Moreover, the genomic cluster responsible for the lipopeptide viscosin production was also identified. Similar to pyoverdin, this compound displays remarkable biocontrol properties against *Aspergillus fumigatus* [72].

The exclusive soluble compounds detected solely within the MCal1 strain encompass bacillomycin D and fengycin. Within the category of lipopeptides, bacillomycin D, classified among the iturin family, stands out as an antifungal secondary metabolite. This compound operates by impairing the cell wall and membrane of fungal hyphae and spores, causing disruptions within the cytoplasm and organelles, resulting in exudation and empty cavities within the cell structure [73]. Similarly, fengycin, another potent antifungal compound potentially produced by MCal1, constitutes a lipopeptide complex specifically targeting filamentous fungi. Notably, it exhibits an ineffectiveness against yeasts and bacteria [74].

In order to check that the strains were indeed also able to inhibit fungal growth through soluble compounds, we carried out experiments in LB agar and Vogel agar culture media supplemented with cell-free supernatants of the BJa3 and MCal1 strains. The results shown in Figure 3 indicate that the strains BJa3 and MCal1’s supernatants contained diffusible compounds that led to the inhibition of the growth of *A. flavus* when the bacteria were cultivated alone in LB broth compared to the negative control (*A. flavus* cultivated alone) (Figure 3).

As the two strains proved to be capable of producing diffusible compounds with antimicrobial activity when cultivated alone, we wondered if they were also able to produce them when cultivated in the presence of the fungus in liquid medium, a condition that should induce a metabolic response in bacterial metabolism. As shown in Figure 3, the growth of *A. flavus* was similarly inhibited both in cell-free supernatants of BJa3 and MCal1 cultivated alone or with *A. flavus* after 24 h of incubation. Thus, in both conditions (bacteria cultivated with or without the fungus), fungal growth decreased by approximately 60% compared to the negative control (Figure 3).

Also, we used supernatants of *E. coli* DH5α as control bacteria, in both the conditions of cultivation alone and with *A. flavus*, with no apparent influence on fungal growth, similar to the negative control. Moreover, the impact of LB and Vogel culture media and their combination on bacteria cultivation was also verified. Thus, as observed in Figure 3, none of the controls (*E. coli* DH5α or culture media) hampered the fungus’ growth or morphology.

### 3.4. VOCs Produced by BJa3 and MCal1 Strains Affect the Morphology of A. flavus 

Once we had demonstrated the inhibition of the growth of *A. flavus* by the new *Pseudomonas* strains, we looked into the structural and morphological alterations in *A. flavus* when it was exposed to the volatiles produced by BJa3, MCal1, and DH5α. To achieve this goal, microcultures of *A. flavus* were created and incubated for 48 h before CLSM analysis. As shown in Figure 4, it was possible to observe that all fungal structures were preserved when *A. flavus* was cultivated without bacteria in the microculture (negative control). The biseriate conidial head, with the primary separation into metulae and phialides, which is very characteristic of the *A. flavus* species [44,75], is in early formation and is well characterized and preserved in the control condition (Figure 4A, arrow head). It is also possible to observe that the vesicle and conidia are being formed and their morphology is intact (Figure 4B, arrow head). Conidia of *A. flavus* have relatively thin walls which are finely to moderately roughened. Their shape can vary from spherical to elliptical [75]. Furthermore, the walls of the hyphae are intact, with well-demarcated septa (Figure 4C, arrow head).

Similarly, when *A. flavus* was cultivated in the presence of *E. coli* DH5α, the resulting fungal structures were similar to the negative control, with the formation of the conidial head and conidia (Figure 4D, white square). The hyphae were also intact and had well-defined septa (Figure 4E,F, arrow head). Thus, according to these results, *E. coli* does not have an impact on the morphology of *A. flavus* or interfere with the development of its vegetative and reproductive structures. 

On the other hand, exposing the fungus to the *Pseudomonas* volatiles caused changes in the cell wall and hyphal germination of *A. flavus* (Figure 4G–L). Accordingly, after 48 h of exposure to the BJa3 isolate, the fungus exhibited signs of late germination, including thinner and less developed hyphae (Figure 4G, white rectangle). Additionally, the hyphae had a scaly, vacuolated appearance without septa (Figure 4H, arrow head and Figure 4I). Similar alterations in the structures of the cell wall and germination were found in *A. flavus* in the presence of the MCal1 strain. In addition to delayed germination (Figure 4J, arrow head), disorganization and destruction of the hyphal structure, as well as internal cell darkening, were also seen, resulting in abnormal fungal cell growth (Figure 4K, arrow head and Figure 4L, white rectangle). Comparable alterations in fungal morphology due to the presence of volatile compounds produced by bacteria were found in the literature [20,76,77,78,79]. For example, some VOCs have been reported to cause structural changes to the cell envelope or even damage the cell wall in the *Thielaviopsis ethacetica* fungus [80]. In this study, the authors proposed that the overexpression of ABC transporter proteins may be related to their role in osmo-adaptation and the maintenance of cell integrity and survival in response to undesirable changes occurring to the cell [20,76,78]. 

Similarly, actinomycete volatiles may alter the morphology of conidiophores and hyphae in a number of fungi [77] and prevent *Cladosporium* spp. conidiophores from germinating [79], as observed in our work (Figure 4G,J). According to Hernandez-León et al. (2015), some *Pseudomonas fluorescens* strains produced VOCs that prevented the growth of the phytopathogenic fungus *Botrytis cinerea* [17]. Analogous to this, *Agrobacterium tumefaciens* and *Agrobacterium vitis* were both inhibited by the VOCs produced by the *P. fluorescens* strain B-4117 [81]. Furthermore, it is essential to note that many current commercially used fungicides also have the membrane and cell wall as their primary targets [82,83].

### 3.5. Strains BJa3, MCal1, and DH5α Produce VOCs That Are Species-Specific and Pseudomonas Strains Produce Compounds with Antifungal Potential

In order to identify which volatiles were produced by microorganisms, the volatilome profile of the *Pseudomonas* strains, *E. coli* DH5α, and *A. flavus* were analyzed by SPME/GC–MS. Although we had a total of 606 compounds, we considered the 47 VOCs that were annotated by GNPS and had statistically significant differences in the spectra peak area in at least one group according to the Kruskal–Wallis test (a = 0.05) (Appendix A). From them, fourteen, fourteen, and twenty VOCs were detected in the volatilomes of BJa3, MCal1, and DH5α, respectively. On the other hand, four, one, and three VOCs were detected exclusively in the volatilomes of the bacteria cultivated in the presence of the fungus (i.e., BJa3 + *A. flavus*, Mcal1 *+ A. flavus* and DH5α *+ A. flavus*, respectively) (Figure 5A). As can be observed in the Venn diagram (Figure 5A), some compounds were shared by the three isolates. These compounds comprise several chemical classes, including alcohols, ketones, benzenoids, alkenes, and others with more complex composition (Appendix A). This approach allowed us to identify a greater variety of compounds produced by our isolates in comparison to the literature, which reports the production of nine to thirty VOCs by *Pseudomonas* strains in different studies [17,64,80,84,85,86].

To elucidate the discriminatory patterns within the volatilomes of the bacterial isolates under investigation, principal component analysis (PCA) was conducted (Figure 5B). As anticipated, bacterial strains cultivated independently exhibited clustering proximal to those cultivated concurrently with the fungus. Notably, the *Pseudomonas* isolates BJa3 and MCal1 demonstrated a closer spatial arrangement in the PCA plot compared to their juxtaposition with *E. coli* DH5α (Figure 5B). Intriguingly, the volatile profile of the fungus cultivated in isolation manifested the most distinctive characteristics among the samples examined.

A heatmap (Figure 6) was generated to show the variations in the relative abundance of selected annotated VOCs produced by antagonist bacteria under different culture conditions. As shown in Figure 6, a clear difference in VOC profiles was observed in this analysis. For example, it was possible to observe that there was a clustering between BJa3 and Bja3 *+ A. flavus* and also between Mcal1 and Mcal1 *+ A. flavus*, as expected. In these cases, it is possible to see that the production of VOCs was similar, although the amount and presence of some compounds is altered. An exception is for *E. coli* DH5α, in which the VOCs produced by the bacteria cultivated separately did not cluster with the VOCs produced by *E. coli + A. flavus.* Although several reasons may have contributed to this result, we suggest that *E. coli*, while sensing the presence of *A. flavus*, is not capable of producing VOCs that alter the growth and production of fungal secondary metabolites when fungus grows together with the bacteria. Thus, it is possible to observe in Figure 6 that the VOCs produced by *E. coli* DH5α co-cultivated with *A. flavus* were grouped with the *A. flavus* volatilome. It is interesting to note that this finding corroborates previous results, in which *E. coli* DH5α does not interfere with fungal growth [20]. Also, it is worth noting that our study showed that co-cultures of bacteria and fungi have emerging volatile metabolomic properties. Therefore, the metabolome of the co-culture cannot be predicted from the sum of its constituents. 

Considering the chemical nature of the compounds annotated in this study, it is possible to notice that some classes of compounds are common among *Pseudomonas* isolates (Figure 6). Moreover, some of these classes have already been validated in the literature for their biocontrol activity, such as alcohols [80,87,88,89], the unsaturated aliphatic hydrocarbons (i.e., 1-heptoxydecane) that were shown to inhibit the growth of several pathogenic fungus [80,90,91], and ketones (i.e., tridecan-2-one) that were confirmed as a bioactive compound against *Fusarium oxysporum* [92]. Furthermore, the same volatile compound may have opposite effects on different target organisms. For example, volatile sulfur compounds such as dimethyl disulfide, dimethyl trisulfide, and dimethyl tetrasulfide can strongly inhibit fungus (*Rhizoctonia solani* and *Fusarium culmorum*) and oomycete (*Pythium ultimum*) growth while promoting the development of some bacteria (*Pseudomonas* sp.) [93,94,95] and some plants [96].

The main mechanism underlying the antifungal properties of VOCs involves the disruption of fungal cell wall and membrane structures, resulting in intracellular lysate leakage and the induction of oxidative stress [97]. This phenomenon is facilitated by the ability of VOCs to permeate fungal cells via hydrogen bonding interactions. Consequently, the forces generated during this interaction perturb the aqueous environment of cell membranes, thereby interfering with their cellular physiology and functionality [97]. Thus, *B. cinerea* treated with VOCs derived from *Streptomyces globisporus* shows excessive vesiculation, thickened walls and retracted membranes [98]. In addition, *Trichoderma* sp., *Phoma* sp., and *Colletotrichum* sp. Exposed to VOCs from *Chromobacterium vaccinii* show extensive morphological abnormalities, such as swollen hyphal cells, vacuolar deposits, and cell wall alterations [99]. This action corroborates our findings in the microscopy of fungi exposed to VOCs.

Several VOCs exhibit a direct targeting mechanism towards fungal cell membranes, thereby increasing membrane permeability and inducing cell leakage. Notably, certain VOCs such as nonanoic acid, synthesized by *Pseudomonas* sp., have been identified as enhancing cell membrane fluidity. This phenomenon is attributed to the alteration of membrane protein conformation, consequently leading to the leakage of intracellular contents [100]. Furthermore, Bergsson et al. (2001) [101] have elucidated the destructive effects of decanoic acid on *Candida albicans* cell membranes, culminating in the release of cytoplasmic contents.

Zhang et al. (2020) [102] found that, in addition to morphological changes in the hyphae and the destruction of the cell membrane, there was a significant accumulation of ROS in *Ceratocystis fimbriata* cells after exposure to VOCs from *Pseudomonas chlororaphis*. Thus, oxidative stress-induced mitochondrial dysfunction and decreased ATP levels inhibited the growth of *C. fimbriata*. In addition, organic acids such as decanoic acid produced by *S. cerevisiae* significantly decrease intracellular ATP levels and inhibit the growth of *B. cinerea*, possibly through mechanisms related to energy metabolism [103]. In general, VOC-induced ROS accumulation and oxidative stress lead to the inhibition of the growth of pathogenic fungi.

## 4. Conclusions

In this study, we explored the efficacy of soil *Pseudomonas* isolates BJa3 and MCal1 as biocontrol agents against the phytopathogenic fungus *A. flavus*. Our findings revealed the significant antifungal activity of both the BJa3 and MCal1 *Pseudomonas* strains, effectively inhibiting the mycelial growth of *A. flavus*. In summary, our work underscores the potential of *Pseudomonas* species to be used as biocontrol agents against phytopathogenic fungi through the production of both diffusible and volatile compounds. Future research endeavors should delve into elucidating the specific mechanisms underlying the antifungal activity of these compounds and exploring their applications in sustainable crop protection practices.

## Figures and Tables

**Figure 1 biotech-13-00008-f001:**
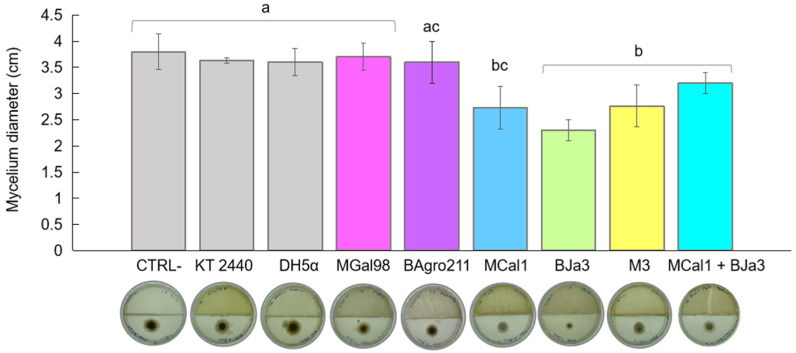
Effect of bacterial VOCs on *A. flavus* growth inhibition after 2 days of co-culture. Inhibitory effect of bacterial VOCs from monocultures and mixtures of BJa3 with MCal1. Controls correspond to *A. flavus* without the bacterial inoculum, *E. coli* DH5α and *Pseudomonas* KT 2440. Different letters indicate significant differences between treatments. Standard errors are indicated by vertical lines (Tukey’s test, *p*-value < 0.05).

**Figure 3 biotech-13-00008-f003:**
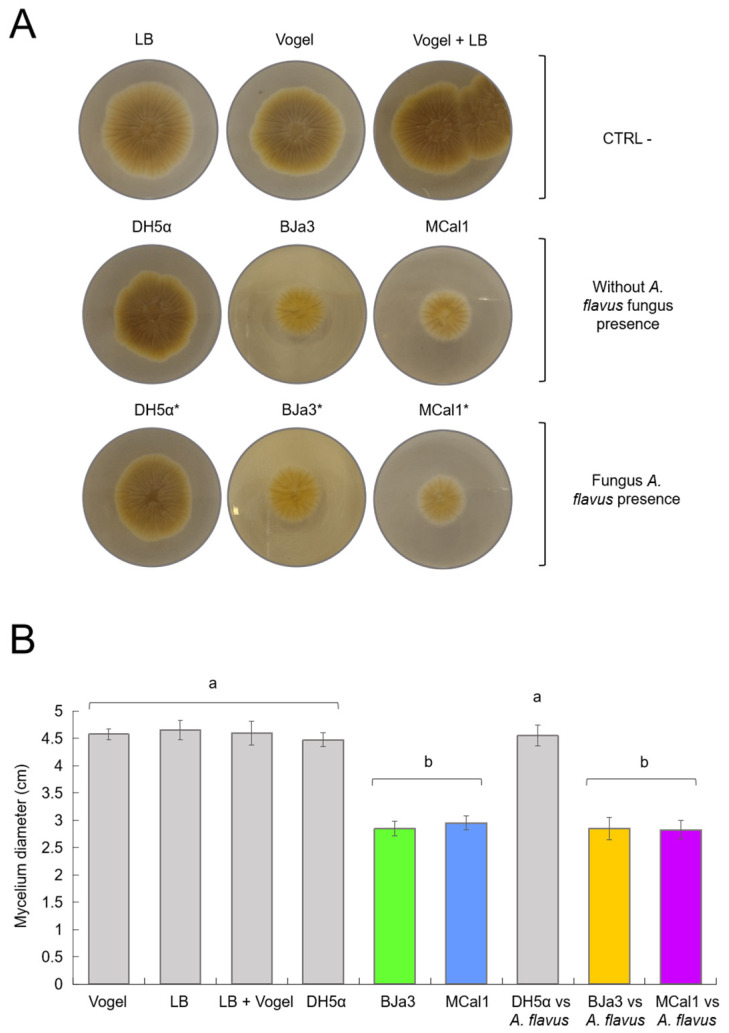
Effect of bacterial-diffusible compounds on *A. flavus* growth inhibition after 2 days of co-culture. (**A**) Controls (CTRL-) corresponding to *A. flavus* without the bacterial inoculum, using Vogel, LB, and Vogel + LB medium, respectively. Effects of diffusible compounds in the absence of fungi with the bacteria *E. coli* DH5α and *Pseudomonas* BJa3 and MCal1, respectively, on the mycelial growth of the fungus *A. flavus* are shown in the second line. Effects of diffusible compounds on the mycelial growth of the fungus *A. flavus* from the bacteria *E. coli* DH5α and *Pseudomonas* BJa3 and MCal1, respectively, when cultivated in the presence of the fungus in liquid medium are represented in the third line. (**B**) Inhibitory effect of soluble bacterial compounds from the 2 selected isolates on the growth of *A. flavus* mycelium. The asterisk (*) indicates the condition of co-culture of bacteria and fungus in the inoculum. Different letters indicate significant statistical differences between treatments. Standard errors are indicated by vertical lines (Tukey’s test, *p*-value < 0.05).

**Figure 4 biotech-13-00008-f004:**
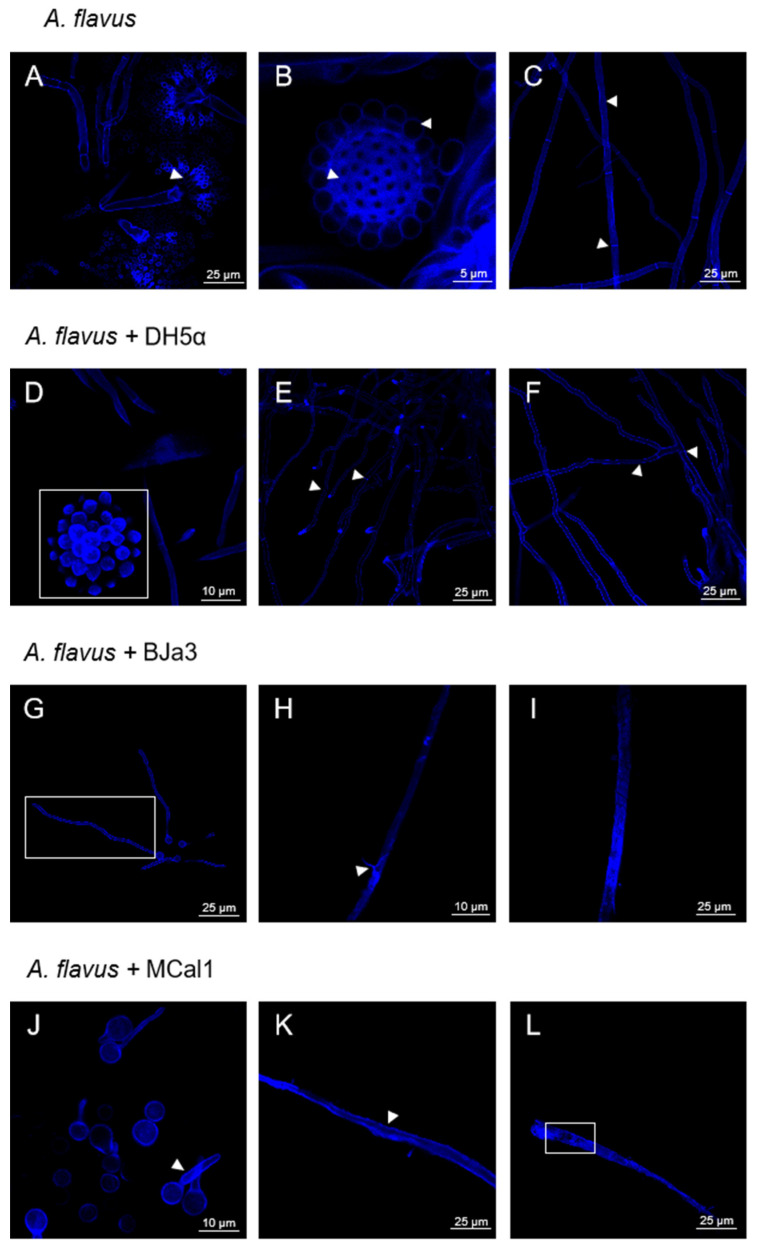
Representative confocal laser scanning microscopy of the mycelium of *A. flavus*. (**A**) Conidial head of *A. flavus*, with phialides highlighted by the arrow head. (**B**) Vesicle and conidia structures are intact and developed, as highlighted by the arrow heads. (**C**) Developed hyphae with demarcated septa, shown by the arrow heads. Images (**A**–**C**) are fungal structures preserved in the absence of bacteria. (**D**) Conidial head with developed conidia, as highlighted by the white square. (**D**,**F**) Preserved and intact hyphae, with septa and intact hyphae structure, as indicated by the arrowheads. Images (**D**–**F**) are fungal structures preserved in the presence of volatiles from *E. coli* DH5α. (**G**) Fungus exhibits signs of late germination, such as thinner and less developed hyphae, delimited by white rectangle. (**H**,**I**) Hyphae with a scaly appearance, as highlighted by the arrowhead and without septa. Images (**G**–**I**) are fungal structures damaged in the presence of volatiles from strain BJa3. (**J**) Late germination of hyphae. (**K**) Hyphae with internal cell darkening, as highlighted by the arrow head. (**L**) Hyphae with internal disorganization and absence of septa, as demarcated by the white rectangle. Images (**J**–**L**) are fungal structures damaged in the presence of volatiles from strain MCal1.

**Figure 5 biotech-13-00008-f005:**
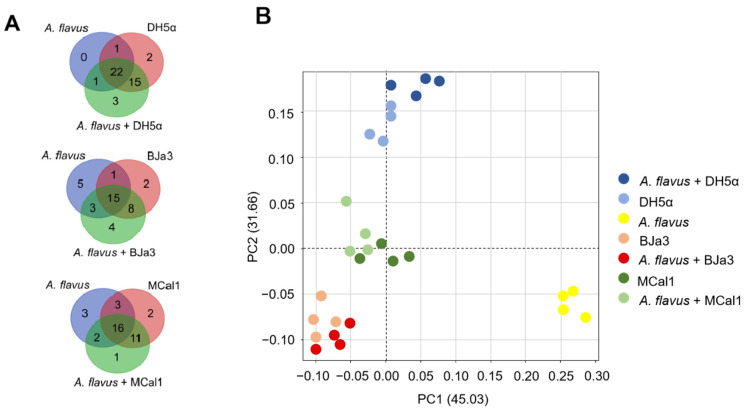
Analysis of the volatilomes produced by the bacteria BJa3, MCal1, *E. coli* DH5α and fungus *A. flavus.* (**A**) Venn diagram considering only the 47 VOCs that have statistically significant differences in spectra peak area in each group. (**B**) PCA of the compounds annotated in the volatilome of the bacteria and fungus.

**Figure 6 biotech-13-00008-f006:**
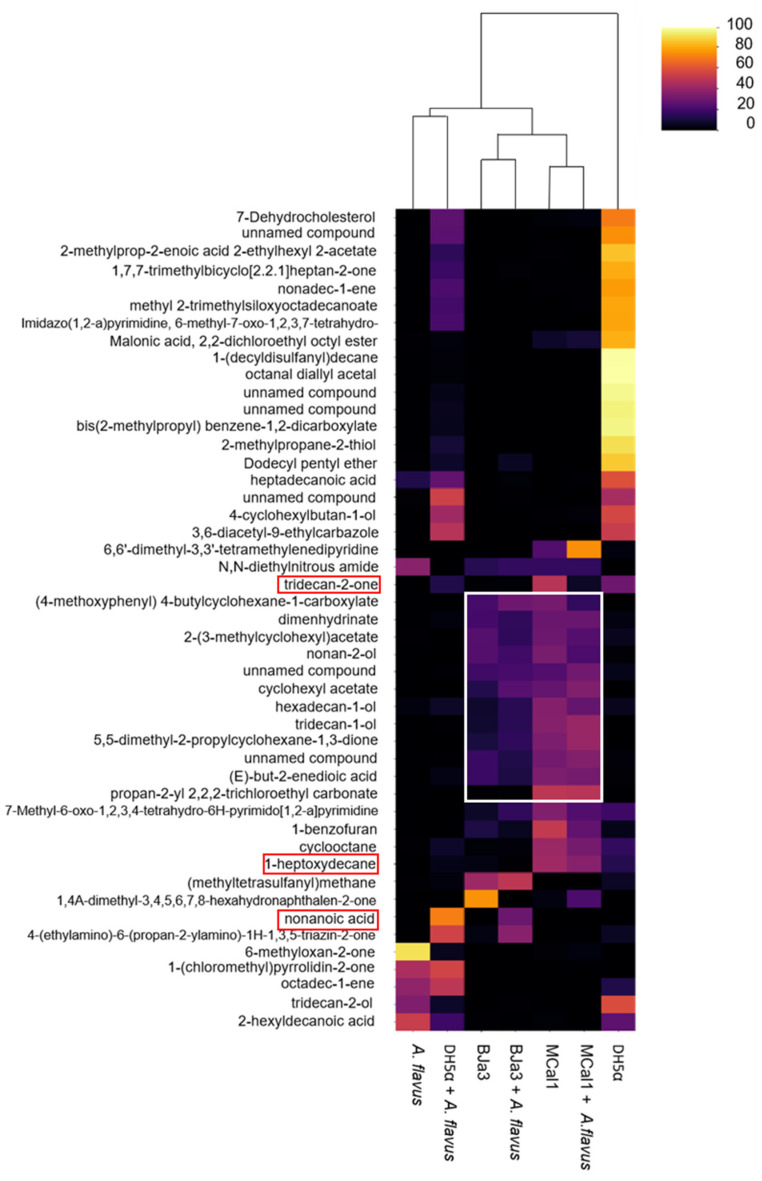
Heatmap of the compounds annotated in the volatilome of the bacteria and fungus. Columns represent each sample condition. Rows represent the different VOCs evaluated. The color code indicates the abundance of each compound (scale from black, low abundance, to yellow, high abundance). The white rectangle indicates the set of volatile compounds produced by the BJa3 and MCal1 bacteria of interest and through the interaction between the bacteria and the *A. flavus* fungus. The red rectangles show some volatile compounds whose antifungal action has already been described in the literature. The columns were clustered based on Euclidean distance from the peak area average of each sample.

## Data Availability

BJa3 and MCal1 genome sequence data were deposited in the NCBI GenBank database under accession numbers JAOXMC000000000 and JAKUMP000000000, respectively (Appendix A).

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
