# Peer review of "Novel Pseudomonas Species Prevent the Growth of the Phytopathogenic Fungus Aspergillus flavus"

_biotech, 2024, doi:10.3390/biotech13020008_

Round 1

Reviewer 1 Report

Comments and Suggestions for Authors

The paper discusses the growing interest in utilizing microbial volatile organic compounds (VOCs) as eco-friendly alternatives to combat phytopathogens. It identifies two Pseudomonas strains capable of inhibiting the mycelial growth of the phytopathogenic fungus Aspergillus flavus, which poses a threat to sugarcane and maize crops. The study highlights the production of 47 distinct VOCs by these bacterial strains, with certain compounds showing promising antifungal activity. Confocal microscopy reveals significant morphological alterations in fungal mycelia exposed to these VOCs, emphasizing the potential of the identified Pseudomonas strains for fungal biocontrol in agricultural crops.

Providing detailed methodologies and experimental procedures

Further elaboration on the potential mechanisms underlying the antifungal activity of identified VOCs and their interaction with fungal physiology would enrich the discussion section.

While the study demonstrates promising results in vitro, future research should focus on field trials to evaluate the efficacy of the identified Pseudomonas strains under realistic agricultural conditions.

Discussing the potential limitations and challenges associated with the practical application of microbial VOCs in agricultural settings would provide a more comprehensive perspective on the feasibility and scalability of this approach

Overall, the paper presents valuable insights into the utilization of microbial VOCs for sustainable agriculture and lays the groundwork for future research in the field of biocontrol strategies against phytopathogens. With further investigation and validation, the identified Pseudomonas strains and their associated VOCs hold significant promise for enhancing crop protection and promoting environmentally friendly agricultural practices.

Reviewer 2 Report

Comments and Suggestions for Authors

This paper investigates the potential of two Pseudomonas strains, BJa3 and MCal1, as biocontrol agents against the phytopathogenic fungus Aspergillus flavus. The authors conducted a comprehensive study to evaluate the antifungal activity of these bacterial strains.

- The manuscript is undoubtedly interesting but requires some minor revisions:

1.     Figure 1: A and B data are largely duplicated. The second graph introduces just one additional column to the first graph's data. Combining these two graphs into one might be beneficial.

2.     In Figure 3.A, second line - “without fungus presence from the bacteria E. coli and Pseudomonas” - what does this mean? Which fungus is being referred to?

3.     Line 533 - Line 533 - Pseudomonas, and line 545 - A. flavus should be italicized.

The main question, however, is what happens to A. flavus mycelium next? Why are the results limited to just two days of incubation? Wherein, complete destruction of the fungus has not been achieved, but only inhibition / slowdown of growth (up to 50%). How detrimental is this damage to the fungus? Can it recover and resume growth? Or are these irreversible changes that will eventually lead to the fungus's demise? This needs to be tested to assess the effectiveness of these bacteria as promising agents for fungal biocontrol.

Reviewer 3 Report

Comments and Suggestions for Authors

1. Certain grammatical errors need to be addressed. For eg. line 39 needs to be corrected. Line 430.

2. In materials and methods, authors should provide a composition of the media used.

3. Authors should include references from 2023 also.

4. The conclusion should be concise and informative. Authors should reduce the length of the conclusion.
